# Research on Path-Planning Algorithm Integrating Optimization A-Star Algorithm and Artificial Potential Field Method

**Lisang Liu** [1,2] , **Bin Wang** [1,2,*] and **Hui Xu** [1,2]

1  School of Electronic, Electrical Engineering and Physics, Fujian University of Technology,
   Fuzhou 350118, China
2  National Demonstration Center for Experimental Electronic Information and Electrical Technology Education,
   Fujian University of Technology, Fuzhou 350118, China
*  Correspondence: 2201905138@smail.fjut.edu.cn

**Abstract:** A fusion pathfinding algorithm based on the optimized A-star algorithm, the artificial potential field method and the least squares method is proposed to meet the performance requirements of path smoothing, response speed and computation time for the path planning of home cleaning robots. The fusion algorithm improves the operation rules of the traditional A-star algorithm, enabling global path planning to be completed quickly. At the same time, the operating rules of the artificial potential field method are changed according to the path points found by the optimal A-star algorithm, thus greatly avoiding the dilemma of being trapped in local optima. Finally, the least squares method is applied to fit the complete path to obtain a smooth path trajectory. Experiments show that the fusion algorithm significantly improves pathfinding efficiency and produces smoother and more continuous paths. Through simulation comparison experiments, the optimized A-star algorithm reduced path-planning time by 60% compared to the traditional A-star algorithm and 65.2% compared to the bidirectional A-star algorithm path-planning time. The fusion algorithm reduced the path-planning time by 65.2% compared to the ant colony algorithm and 83.64% compared to the RRT algorithm path-planning time.

**Keywords:** A-star algorithm; artificial potential field method; least squares method; path planning

## 1. Introduction

Thanks to the rapid development of artificial intelligence technology, cleaning robots are now being introduced into ordinary households. One of the key parts of a cleaning robot is the path planning of the cleaning area [1,2]. Path-planning technology involves following certain pathfinding rules in an environment with obstacles to obtain a collision-free path from the starting point to the target point that satisfies the evaluation metrics [3,4]. As the complexity of the working environment of mobile robots continues to increase, it also places higher demands on path-planning techniques. Depending on the environment in which the robot works and the work requirements, it can be divided into global path planning and local path planning [5,6]. Global path planning can be divided into graph-based search algorithms, sampling-based planning algorithms, biomimetic-based algorithms, neural network algorithms, etc. Local path planning is divided into the dynamic window method, time-elastic band method, artificial potential field method, etc.

In 1959, Dijkstra, a Dutch scientist, was the first to propose an algorithm to solve the single-source shortest path problem, which was one of the earliest global path-planning algorithms [7]. The algorithm is centered on the starting point using a breadth-first search strategy to continuously expand outwards and then continuously search for the shortest path between the starting point and each expanded node in the map until it finds the target node, completing the path planning. In 1968, P. Hart, N. Nilsson and B. Raphael first proposed the heuristic A-star algorithm to solve the global path optimal problem [8]. The traditional A-star algorithm starts from the starting point and calculates the cost of

moving the current node to the starting point and the ending point under the constraint of the evaluation function and extends radially to the target point. When an obstacle is encountered, it returns to the vicinity of the starting point to resume pathfinding and repeats until the target point is reached. Therefore, the algorithm generates a large number of useless nodes to be computed in the process of application, which leads to problems such as too much computation, too much memory occupation and too long pathfinding time. X. Zhang [9] optimized the algorithm by introducing a time factor to the A-star algorithm and combining it with a time window and priority strategy. Although this method reduces the number of turns and improves the efficiency of the system, it greatly increases the computational effort, and the chosen obstacle avoidance strategy tends to cause the algorithm to fall into a dead loop.

The artificial potential field method was proposed by Khatib in 1986. The idea is to simulate the environment in which the mobile robot is located as the "gravitational force" in physics, called a virtual potential field [10]. A virtual artificial potential field is formed by the repulsive field of an obstacle and the gravitational field of the target location, in which the mobile robot is influenced by the potential field to automatically search for a suitable collision-free path. As the robot moves, the potential field it is subjected to varies continuously, along a gradient from the repulsive field of a particular obstacle or the gravitational field of a target point alone. Due to its real-time nature, the artificial potential field method has been applied to the field of dynamic obstacles by many scholars. However, for scenarios where multiple obstacles exist at the same time, the problem of becoming caught in local minima that cannot be dislodged and the phenomenon of oscillation near the target point easily occur. Y. Wang et al. [11] addressed the problem that the potential field method tends to fall into local optima by improving the gravitational formulation of the traditional potential field method by adding new variables and also expanding the obstacle to a circular obstacle. The optimized algorithm solves the problem of the manual potential field method not being able to avoid large obstacles, greatly reduces the scanning time and reduces the working cost of the mobile robot. However, in an environment with irregular obstacles, it easily divides the passable paths between obstacles into obstacle regions, thus failing to obtain the optimal path. When the map is updated quickly, its real-time obstacle avoidance capability will be greatly reduced. H. Liu et al. [12] introduced the idea of fuzzy control into the path planning of mobile robots, dividing the environment in which the robot moves into two parts: a global safety region and a local danger, according to the location of obstacles and their influence range. In safe areas, the artificial potential field method acts on the robot to guide it towards the target; in dangerous areas, the artificial potential field method is combined with fuzzy control to guide the robot to avoid obstacles and move towards the target. The precise control of the deflection angle of the mobile robot effectively reduces the problem of unreachable targets and local minima. However, the fusion algorithm is limited in its application, and in dynamic environments, it relies mainly on the artificial potential field method, which does not practically solve the shortcomings of the artificial potential field method in dynamic environments.

This paper first optimizes the structure of the traditional A-star algorithm, then optimizes the potential field method by adding an intermittent point search strategy and constructing an intermittent point judgment function, and then combines the artificial potential field method and the least squares method to propose a fused path-planning algorithm. On the one hand, the structure of the traditional A-star algorithm is optimized to improve its speed in global path planning, and on the other hand, the artificial potential field method is optimized to solve the problem of the potential field method tending to fall into local optimality. The fusion algorithm overcomes the drawbacks of the original algorithm well and improves the efficiency and success rate of path planning. The paper concludes with a comparative simulation analysis of the optimized A-star algorithm and the traditional and bidirectional A-star algorithms for different starting points and different map environments. The fusion algorithm is analyzed and compared with the ant colony

algorithm and the RRT algorithm. The simulation comparison and data analysis confirm the fast, robust and advanced nature of the optimized fusion algorithm.

## 2. Global Path Planning

### 2.1. Traditional A-Star Algorithm

The A-star algorithm is a heuristic search algorithm for finding optimal paths in static obstacle environments [13]. It combines the advantages of Dijkstra's algorithm to find the shortest path well and the heuristic search algorithm breadth-first search (BFS) to search upwards at the most probable places first [14], to which a cost evaluation function is added to find the optimal path point. The principle is shown in Figure 1.

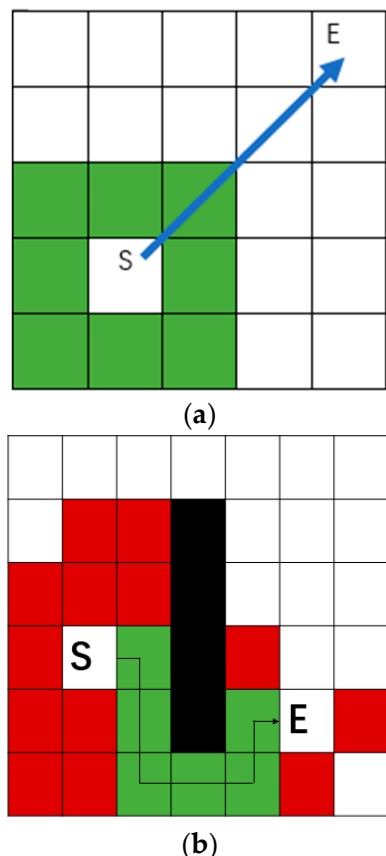

**Figure 1.** Schematic diagram of the traditional A-star algorithm. (**a**) Conventional A-star algorithm pathfinding in the absence of obstacles; (**b**) conventional A-star algorithm pathfinding in the presence of obstacles.

As shown in Figure 1a, S is the starting point and E is the end point. If the path encounters an obstacle, the algorithm returns to the starting point and searches again until the search width exceeds the width of the obstacle, then continues to search and so on until it reaches the target point. In practice, only a small number of nodes are relevant to the path, but many nodes need to be computed. For this reason, a large number of useless nodes are searched, creating problems such as too many calculations, more useless memory usage and longer pathfinding times. In addition, there are too many turning points in the planned path. The traditional A-star algorithm does not smooth the path, so the planned path turns rigidly, and the robot needs to accelerate and decelerate frequently to perform the turning action when walking, which is not conducive to the robot's path tracking.

The set of optimal path points then forms the optimal path, where the cost evaluation function is as follows:

$$f(n) = h(n) + g(n) \tag{1}$$

where $f(n)$ is the cost function of the current position, $g(n)$ is the actual cost of the mobile robot from the starting point to the current position and $h(n)$ is the estimated cost of the mobile robot from the current position to the position of the target point [15].

For the cost function, the more commonly used metric is the Euclidean or Manhattan distance. The absolute value of the difference between the x-coordinates of two points and the sum of the absolute values of the differences between the y-coordinates of two points is called the Manhattan distance [16]. In this paper, the Euclidean distance is used, i.e.,

$$g(n) = \sqrt{(X_n - X_s)^2 + (Y_n - Y_s)^2} \qquad (2)$$

$$h(n) = \sqrt{(X_t - X_n)^2 + (Y_t - Y_n)^2} \qquad (3)$$

where $(X_n, Y_n)$ is the position of the current point, $(X_s, Y_s)$ is the position of the starting point and $(X_t, Y_t)$ is the position of the target point. The closer the value of the function $h(n)$ is to the actual value, the more efficient and accurate the search will be.

### 2.2. Optimization of the A-Star Algorithm

Based on the shortcomings of the traditional A-star algorithm, the algorithm structure of the A-star algorithm is improved so that when an obstacle is encountered during the pathfinding process, instead of returning to the vicinity of the starting point for a new pathfinding instance, the algorithm defines the node that the mobile robot is currently on as the parent node. The open list is used to store the data of the parent node and the neighboring points with the parent node as the core and to filter the next walkable path point of the mobile robot based on the open list. The close list is defined to store the entire set of walkable path points for the mobile robot. As this paper uses Euclidean distances, the F-value is the distance from the current position of the mobile robot to the target point. The G-value is the distance from the current position of the mobile robot to the starting point. These are the values of $f(n)$ and $g(n)$ at the current point as stated above.

The optimized A-star algorithm sets the evaluation function of the obstacle node in situ to infinity, indicating unreachability, and then finds the best node among the nodes around the parent node as the parent node for the next cycle. This continues until the path found leaves the obstacle node, and then the path search continues forward. The principle is shown in Figure 2.

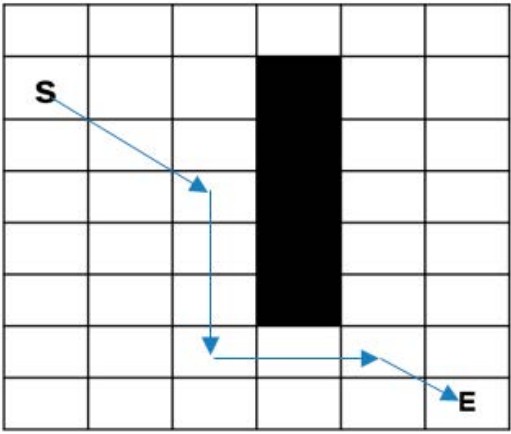

**Figure 2.** Optimization A-star algorithm schematic.

The algorithm steps are as shown in Figure 3:

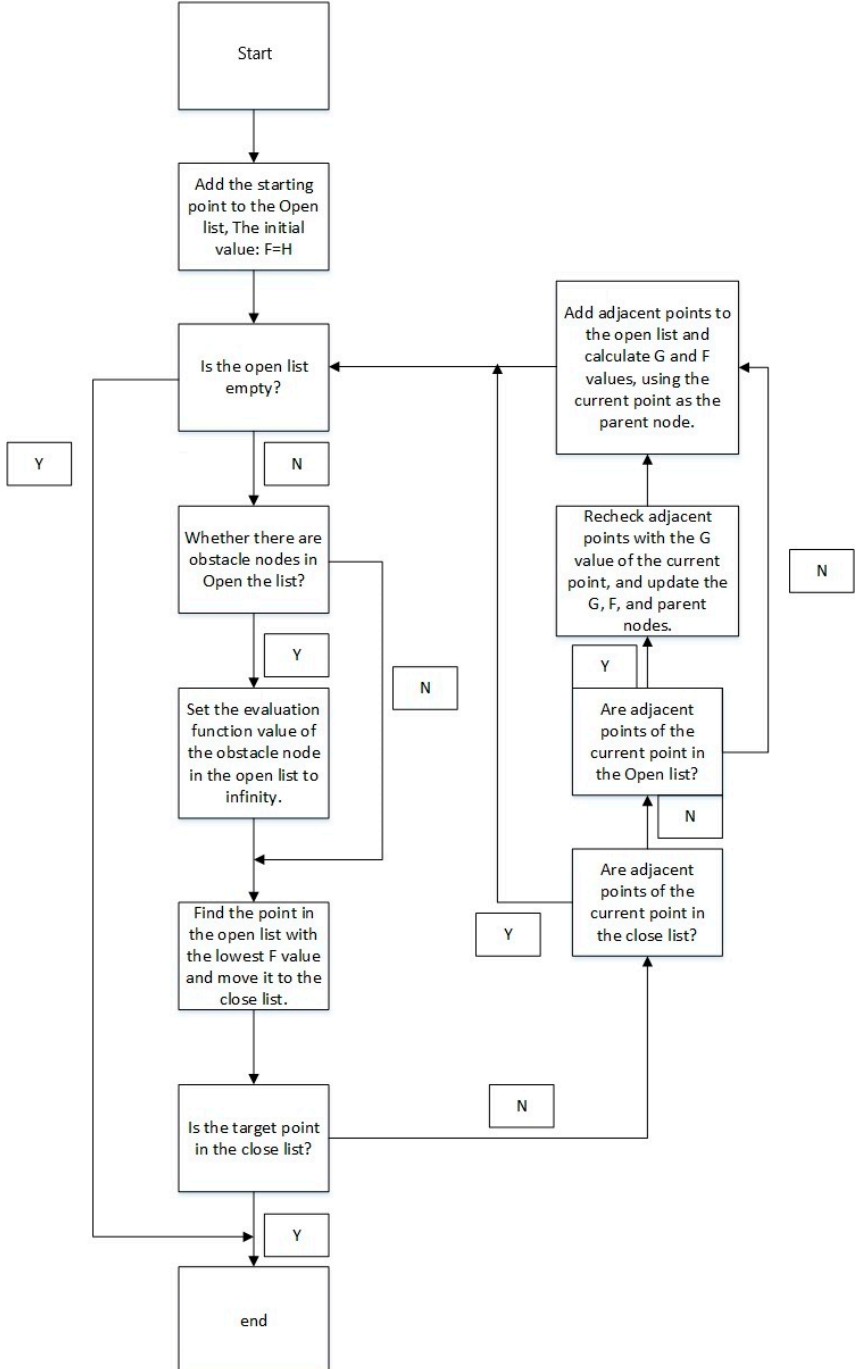

**Figure 3.** Optimization of the A-star calculation hair path-planning process.

The optimized A-star algorithm modifies the path-planning rules of the traditional A-star algorithm. When an obstacle is encountered, the optimized A-star algorithm no longer returns to the vicinity of the starting point to re-run the path planning. When an obstacle is encountered, the optimized A-star algorithm stays put and sets the value of the cost function of the obstacle node from the open list to infinity. The best of the child nodes is then selected as the parent node for the next cycle. Compared to traditional A-star algorithms, the optimized A-star algorithm will reduce the amount of computation and data redundancy, thus reducing path-planning time. As the complexity of the environment increases, the benefits of the optimized A-star algorithm will become more apparent. Detailed simulation comparisons and data analysis will be shown in Section 4.1 of this paper.

### 3. Local Route Planning

*3.1. Artificial Potential Field Method*

The artificial potential field method relies on the repulsive force $F_{rep}(x)$ (shown in Equation (7)), which is directed from the obstacle to the mobile robot, and the gravitational force $F_{aat}(x)$ (shown in Equation (5)), which is directed from the mobile robot to the target point, to construct the gravitational field [17–19]. The gravitational field varies with the distance between the vehicle and the target point. The gravitational field is proportional to the linear distance between the moving vehicle and the target point, as shown below.

$$U_{att}(x) = \frac{1}{2}K\rho^2(P_S, P_E) \tag{4}$$

where $U_{att}(x)$ is the gravitational potential field generated by the target on the mobile robot, $K$ is the coefficient of action of the gravitational field and $\rho(P_S, P_E)$ is the Euclidean distance from the starting point to the endpoint.

The gravitational force is the negative gradient of the gravitational potential field, as follows:

$$F_{aat}(x) = -\nabla U_{att}(x) = -K\rho(P_S, P_E) \tag{5}$$

The magnitude of the repulsive field is inversely proportional to the distance between the mobile robot and the target point, as follows:

$$U_{rep}(x) = \begin{cases} \frac{1}{2}K_{rep}\left[\frac{1}{\rho(P,P_{obs})} - \frac{1}{P_0}\right]^2 & ,\rho(P, P_{obs}) \le P_0 \\ 0 & ,\rho(P, P_{obs}) \ge P_0 \end{cases} \tag{6}$$

where $U_{rep}(x)$ is the repulsive field of the obstacle, $K_{rep}$ is the coefficient of action of the repulsive field, $\rho(P, P_{obs})$ is the Euclidean distance between the mobile robot and the obstacle and $P_0$ is the critical distance of the repulsive force on the obstacle. When the distance $P_0$ between the trolley and the obstacle is greater, the repulsive force on the trolley is zero [20]. Meanwhile, the repulsive force is the negative gradient of the repulsive field, as follows:

$$F_{rep}(x) = -\nabla U_{rep}(x) = \begin{cases} K_{rep}\left[\frac{1}{\rho(P,P_{obs})} - \frac{1}{P_0}\right]\frac{1}{\rho^2(P,P_{obs})} & ,\rho(P, P_{obs}) \le P_0 \\ 0 & ,\rho(P, P_{obs}) \ge P_0 \end{cases} \tag{7}$$

When there are $N$ obstacles on the map, the combined force on them is as follows:

$$F_{sum}(x) = F_{aat}(x) + \sum_{i=1}^{N} F_{rep}(x) \tag{8}$$

where $F_{sum}(x)$ is the repulsive force, $\sum_{i=1}^{N} F_{rep}(x)$ is the combined gravitational force and $F_{sum}(x)$ is the set of repulsive forces.

As shown in Figure 4, the artificial potential field approach to path planning involves the mobile robot following the direction of the combined force $F_{sum}$. The combined force is generated by the combination of multiple repulsive forces $F_{rep}$ exerted by the obstacle on the mobile robot and gravitational forces $F_{aat}$ exerted by the target point on the mobile robot. The repulsive, gravitational and combined forces all follow the rule of vector addition and subtraction. As shown by Equations (5) and (7), the repulsive and gravitational forces change as the position of the mobile robot in the map environment changes. This, therefore, causes the combined forces to change as well. It is the real-time nature of the artificial potential field method that allows the artificial potential field method to be used for local path planning.

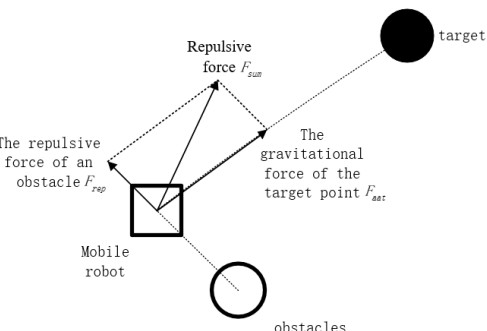

**Figure 4.** Force analysis of robot in artificial potential field.

Due to the pathfinding rules of the potential field method, there is an inherent problem with the traditional artificial potential field approach to path planning; when the robot is moving near the target point, if there are no obstacles near the target point, the gravitational force will be close to zero and the repulsive force should also be close to zero at this point. If there is an obstacle near the target, the robot will be subjected to a large repulsive force, when the gravitational force on the robot at the target is less, which will cause the robot to move away from the target, preventing it from ever reaching it. Secondly, when the robot moves to certain locations on the map, it may be that the combined gravitational and repulsive forces at that location are zero and the robot will fall into a local optimum.

*3.2. Optimization of the Artificial Potential Field Method*

3.2.1. Interruption Point Selection

In this paper, an interruption search strategy is proposed to improve the speed of path planning by the artificial potential field method. The optimized artificial potential field method optimizes the A-star algorithm on the basis of the global path-planning data obtained. The optimized potential field method uses the turning points of the global paths as intermittent points. The turning point of the global path is the place where the path obtained from the global path planning takes a turn. The judgment function is shown in Equations (6) and (7).

$$K_1 = \frac{X_c - X_{c-1}}{Y_c - Y_{c-1}} \tag{9}$$

$$K_2 = \frac{X_{c+1} - X_c}{Y_{c+1} - Y_c} \tag{10}$$

where $(X_c, Y_c)$ are the coordinates of the current point. As the global path-planning data are stored on a stack, $(X_{c-1}, Y_{c-1})$ are specified as the coordinates of the point after the current point. $(X_{c+1}, Y_{c+1})$ are the coordinates of the point before the current point. $K_1, K_2$ are the slope of the line connecting the current point to the two adjacent points before and after it. When $K_1, K_2$ are not equal, the current point is the turning point.

As shown in Figure 5. Starting from point 1, point 3 is the temporary endpoint of point 1, and point 4 is the temporary endpoint of point 2. In the artificial potential field method of smoothing, point 2 is the starting point when the distance from the robot's position to 1 is greater than the spacing from 1 to 2. When it reaches the endpoint, the endpoint is used as the temporary endpoint and the penultimate path point is used as the temporary start point. For this reason, in the manual potential field method for local pathfinding, this paper uses the Manhattan distance for determining whether an intermediate point has been passed. To improve the efficiency of the optimal potential field method and reduce data redundancy, this paper proposes an adaptive number of iterations, which is formulated as follows:

$$I = L * \sqrt{(R_X - T_{EX})^2 + (R_Y - T_{EY})^2} \tag{11}$$

where $L = 10$, and each reference value corresponds to a Euclidean distance of 1. $I$ is the iteration parameter. $(R_X, R_Y)$ are the coordinates of the robot's position. $(T_{EX}, T_{EY})$

are the coordinates of the current temporary target point. Iteration parameters are used in the optimized potential field method when performing path planning. The iteration parameters determine whether the optimized potential field method path planning will reach the target point or not. The Euclidean distance between the current position of the robot and the temporary target point is rounded upwards. When the mobile robot changes its temporary start points and temporary endpoints, the number of iterations is adjusted.

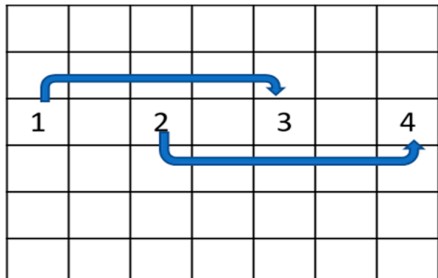

**Figure 5.** Potential Field Method Wayfinding.

### 3.2.2. Least Squares Method

The path obtained by the potential field method will look like a discontinuous path because the interval between the temporary start point and the temporary endpoint is a turning point in the path-planning process of the optimal potential field method. The principle is shown in Figure 6, points 1, 2, 3 and 4 are turning points. Where points 2 and 3 are also turning points. The path discontinuity appears when 2 is the temporary starting point. Therefore, whenever there are turning points in the global path, the most dominant field method will produce path discontinuities after smoothing.

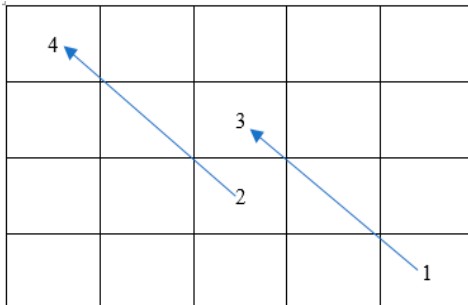

**Figure 6.** Potential field method misalignment principle.

Path fitting is currently more commonly used in the interpolation and least squares methods. The interpolation method is used in environments where the accuracy and reliability of the observed data are high. The interpolation method seeks high accuracy, which results in large data redundancy and can lead to longer path-planning times, resulting in failure to avoid dynamic obstacles in a timely manner. The least squares method is suitable for situations where the observed data already contain unavoidable errors and it is only necessary to reach as close to them as possible. The least squares method is fast and can also fulfill path-planning requirements, so this paper uses least squares for path fitting [21,22].

To solve the path discontinuity problem of the optimal field method, this paper proposes the combination of the least squares method for path fitting, so that the mobile robot can obtain a smooth and continuous path trajectory. The variance between the hypothetical regression results and the actual values is expressed as follows:

$$\phi(x) = a_0 + a_1 x + a_2 x^2 + \cdots + a_k x^k \tag{12}$$

where $a$ is the polynomial's indeterminate coefficient and $x$ is the path point's abscissa. The total distance between each point and this curve is as follows:

$$R^2 = \sum_{i=1}^{n} \left[ y_i - (a_0 + a_1 x_i + a_2 x_i^2 + \cdots + a_k x_i^k) \right]^2 \tag{13}$$

where $n$ is the polynomial's highest order and $k$ is the highest order of the system. We can obtain the following results by deriving the indeterminate coefficients in the regression equation:

$$\begin{bmatrix} 1 & x_1 & \cdots & x_1^k \\ 1 & x_2 & \cdots & x_2^k \\ \vdots & \vdots & \ddots & \vdots \\ 1 & x_n & \cdots & x_n^k \end{bmatrix} \begin{bmatrix} a_0 \\ a_1 \\ \vdots \\ a_k \end{bmatrix} = \begin{bmatrix} y_1 \\ y_2 \\ \vdots \\ y_n \end{bmatrix} \tag{14}$$

The coefficient matrix A and the fitting curve can be generated simultaneously using matrix operation, as shown below.

$$X^* A = Y \Rightarrow A = (X^* X)^{-1} X^* Y \tag{15}$$

Therefore, the algorithmic flow for optimizing the artificial potential field method is as shown in Figure 7:

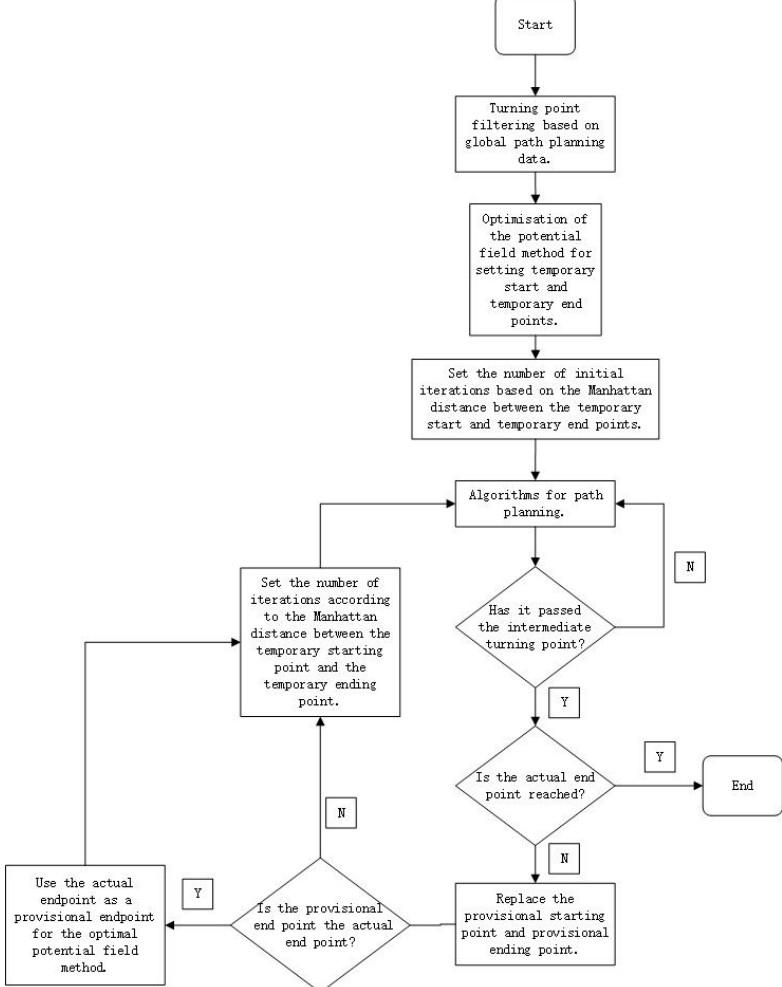

**Figure 7.** Optimization of the potential field method path-planning process.

The optimized potential field method changes the start and end points in pathfinding and solves the problem of falling into local optima and unreachable target points by setting parameters such as step size reasonably. At the same time, the optimized potential field method takes the path obtained from global planning as the general direction, so it also limits the pathfinding range of local paths and avoids crossing between multiple obstacles. At the same time, since the potential field method gives the mobile robot a repulsive force during path planning, pushing the robot away from the obstacle appropriately, the fusion algorithm page solves the problem of narrower paths for the mobile robot to pass through. In addition, as the optimized A-star algorithm can only perform global path planning, moving obstacles may appear on the map when the robot moves, so given the nature of the potential field method updating the map in real time, the potential field method can perform path smoothing while also performing dynamic obstacle avoidance.

## 4. Simulation and Analysis

### 4.1. Comparative Analysis of Optimization Algorithms and Traditional Algorithms

To verify the effectiveness and generalization of the fusion algorithm based on the optimized A-star algorithm and the artificial potential field method proposed in this paper, MATLAB simulations of the traditional A-star algorithm and the optimized A-star algorithm, the traditional potential field method and the optimized potential field method were carried out in simple and complex environments to verify the performance of the optimization algorithm as proposed in this paper.

A simple environment mapping is shown in Figure 8. A raster map of three different environments was constructed in MATLAB, with a map size of 20 × 20 and black x's indicating obstacles. The red boxed points are the calculated path points, the green boxed points are the points to be included in the open list to be checked and the connecting lines are the optimal paths found. From Figure 8, it can be seen that the optimized A-star algorithm can obtain the same path as the traditional A-star algorithm under the same map environment. A comparison of the path-planning times for the 10 groups based on Figure 8a,b is shown in Table 1, where the optimized algorithm reduces the path-planning time by 60% compared to the traditional algorithm. In addition, based on Figure 8b,d,e,f, the optimized A-star algorithm can obtain a feasible path quickly and accurately in the same map environment with different start and end point settings.

**Table 1.** Comparison of path-planning times based on Figure 8a,b (unit: s).

| Time | 1 | 2 | 3 | 4 | 5 | 6 | 7 | 8 | 9 | 10 |
|---|---|---|---|---|---|---|---|---|---|---|
| Traditional A-star algorithm | 0.623 | 0.583 | 0.635 | 0.592 | 0.606 | 0.581 | 0.604 | 0.600 | 0.596 | 0.603 |
| Optimization of the A-star algorithm | 0.368 | 0.225 | 0.208 | 0.190 | 0.218 | 0.175 | 0.179 | 0.172 | 0.170 | 0.174 |

As shown in Figure 9, after the map environment is changed, the optimized algorithm can still meet the path-planning requirements. Compared with the traditional A-star algorithm, the optimized A-star algorithm searches a much smaller range of path points than the traditional algorithm while obtaining the same path in the same map environment. As shown in Figure 9b,d, the optimized A-star algorithm can complete the path-planning requirements in the new map environment with different start and end points replaced. As shown in Figure 9a,b and the path-planning time comparison in Table 2, the path-planning time of the optimized A-star algorithm is reduced by more than 50% compared to the traditional A-star algorithm.

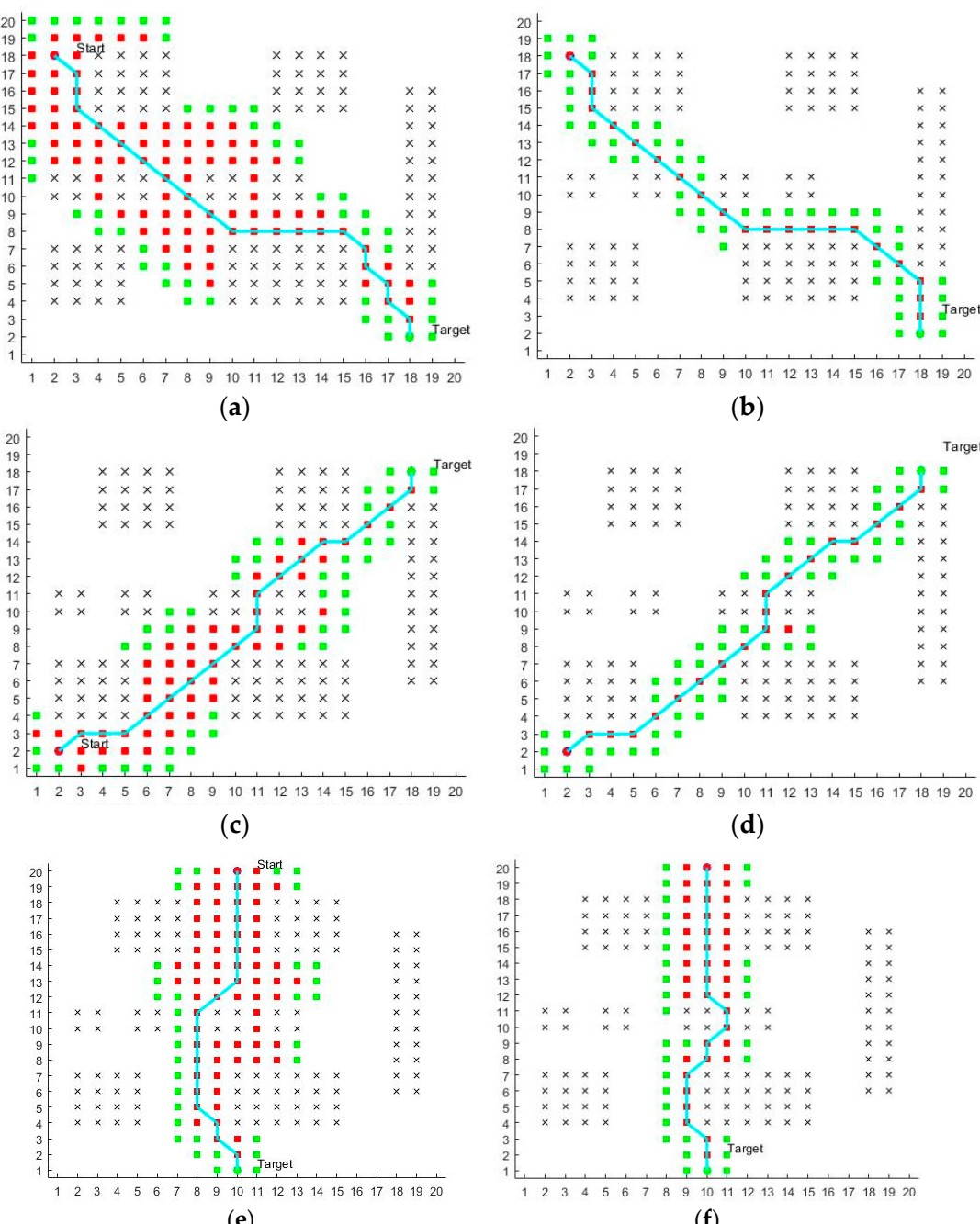

**Figure 8.** Global path comparison. (**a**,**c**,**e**) Traditional A-star algorithm; (**b**,**d**,**f**) optimization of the A-star algorithm.

**Table 2.** Comparison of path-planning times before and after algorithm optimization (unit: s).

| Time | 1 | 2 | 3 | 4 | 5 | 6 | 7 | 8 | 9 | 10 |
|---|---|---|---|---|---|---|---|---|---|---|
| Traditional A-star algorithm | 0.309 | 0.307 | 0.308 | 0.305 | 0.303 | 0.299 | 0.304 | 0.300 | 0.296 | 0.299 |
| Optimization of the A-star algorithm | 0.140 | 0.139 | 0.139 | 0.139 | 0.140 | 0.138 | 0.136 | 0.134 | 0.133 | 0.133 |

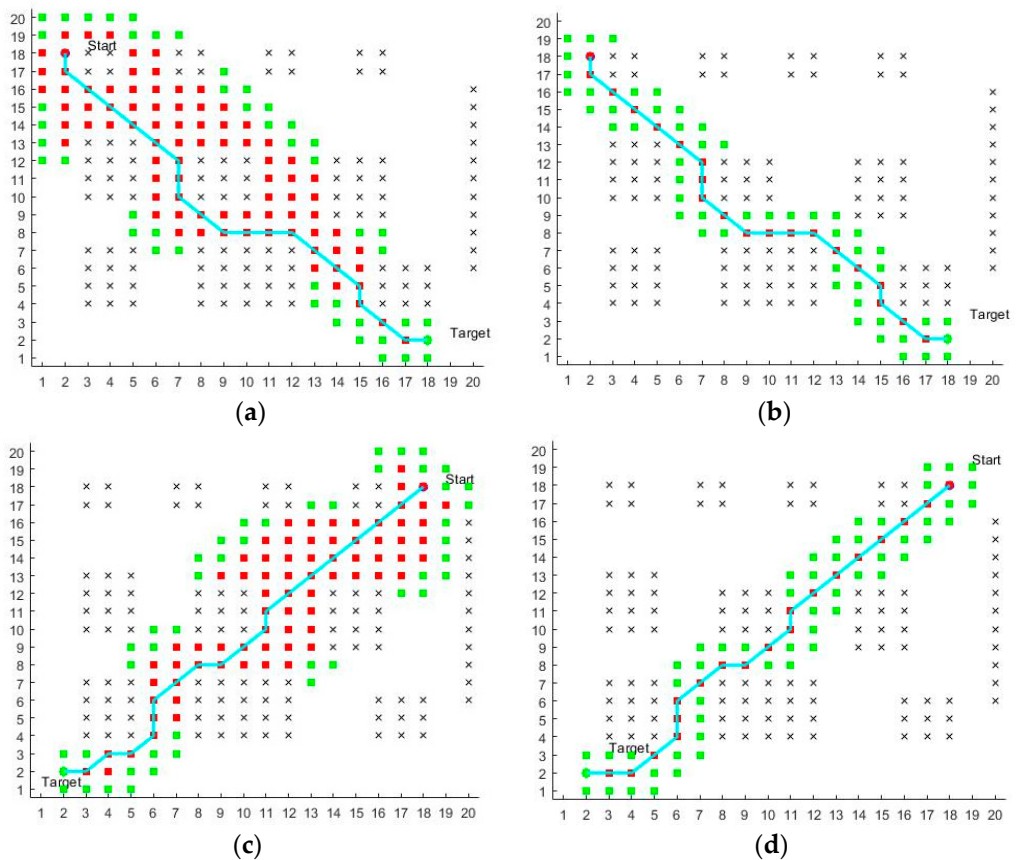

**Figure 9.** Global path comparison for map replacement. (**a**,**c**) Traditional A-star algorithm; (**b**,**d**) optimization of the A-star algorithm.

The complex map is shown in Figure 10, and a 40 × 40 grid map was created in MATLAB. The optimization algorithm is still able to obtain a feasible path when performing path planning in a complex map environment.

Compared with the traditional A-star algorithm, the advantage of less computation of the optimized A-star algorithm is more obvious. A comparison of path-planning times based on Figure 10a,b, as shown in Table 3, shows that the optimized A-star algorithm reduces the pathfinding time by nearly 70% compared to the traditional algorithm.

**Table 3.** Comparison of path-planning times for complex maps (unit: s).

| Time | 1 | 2 | 3 | 4 | 5 | 6 | 7 | 8 | 9 | 10 |
|---|---|---|---|---|---|---|---|---|---|---|
| Traditional A-star algorithm | 2.427 | 2.436 | 2.419 | 2.371 | 2.402 | 2.552 | 2.349 | 2.359 | 2.371 | 2.368 |
| Optimization of the A-star algorithm | 0.683 | 0.732 | 0.703 | 0.689 | 0.691 | 0.782 | 0.704 | 0.699 | 0.685 | 0.692 |

Based on the analysis of the path-planning time and the path accessibility, the optimized A-star algorithm achieves a significant improvement in path accessibility and speed over the traditional algorithm, while ensuring that a complete global path can be obtained. As the complexity of the pathfinding environment increases, the efficiency of the optimized algorithm becomes more pronounced than that of the traditional algorithm. Nevertheless, the optimized A-star algorithm does not solve the problem of insufficient smoothness at path transitions.

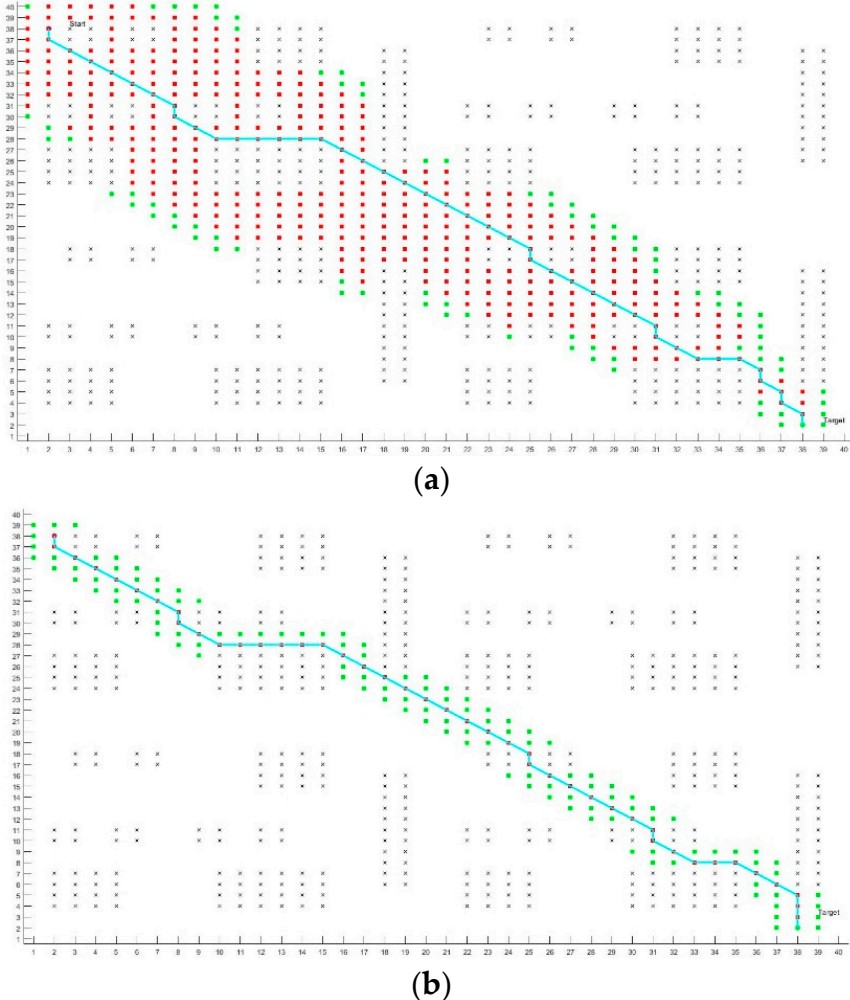

**Figure 10.** Comparison of global path planning for complex map environments. (**a**) Traditional A-star algorithm; (**b**) optimization of the A-star algorithm.

To address the problem of the global path-planning transitions not being smooth enough to facilitate smooth robot tracking, an optimized potential field method is proposed for smoothing, as described in the previous section. The simulation diagram is shown in Figure 11.

Figure 11 also shows that the optimization algorithm is still able to meet the pathfinding requirements when different starting points and different endpoints are set in the same environment. Moreover, from Figure 11b,d, it can be seen that the optimization algorithm can effectively complete the library path planning when the same start and end points are set in different map environments. It can also be seen from Figure 11 that the algorithm has good robustness and generalizability. The comparison between Figures 11 and 12 shows that the smoothing process reduces a large number of inflection points compared to global path planning, thus improving the path-tracking capability of the robot and increasing the movement speed of the mobile robot. In addition, as the repulsive force of the obstacle in the artificial potential field method acts on the cart, it will cause the cart to move away from the obstacle appropriately, making the path of the cart more reasonable and solving the global path-planning problem of walking along the edge of the obstacle.

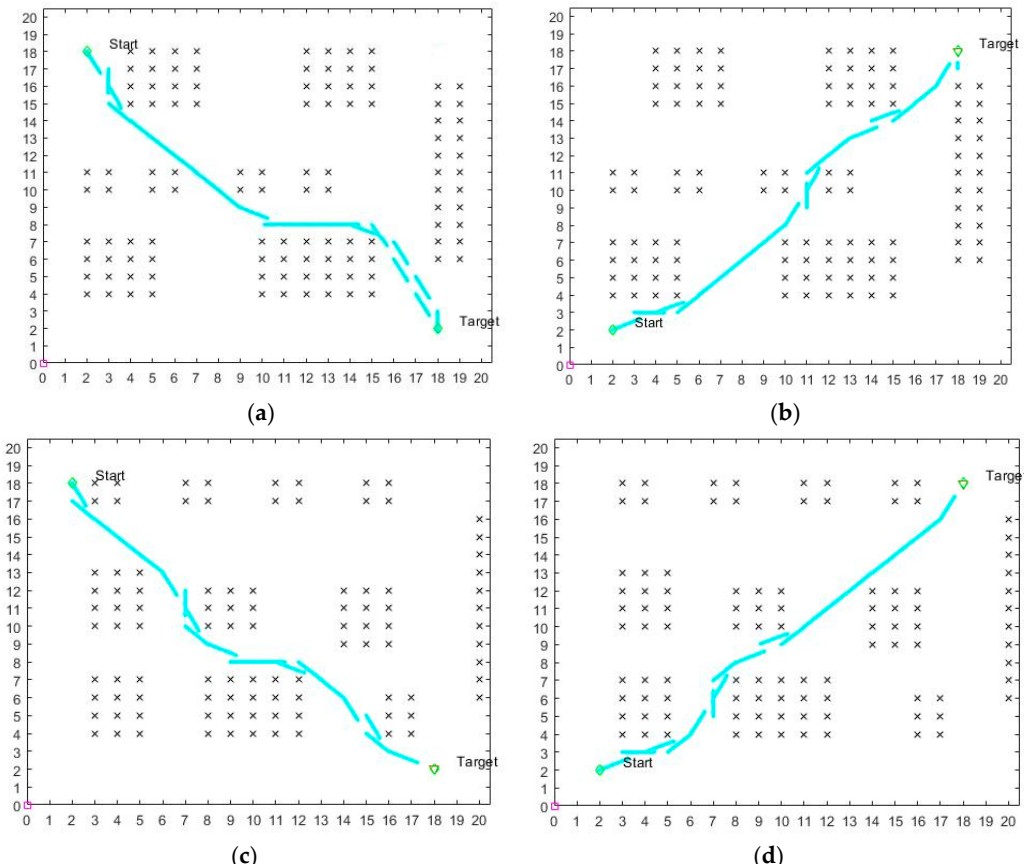

**Figure 11.** Use of the optimized potential field method in different map environments to set different start and end points of the path planning. (**a**,**b**) are route plans for different starting points and different end points in the same environment; (**c**,**d**) are route plans for different starting points and different end points after changing maps.

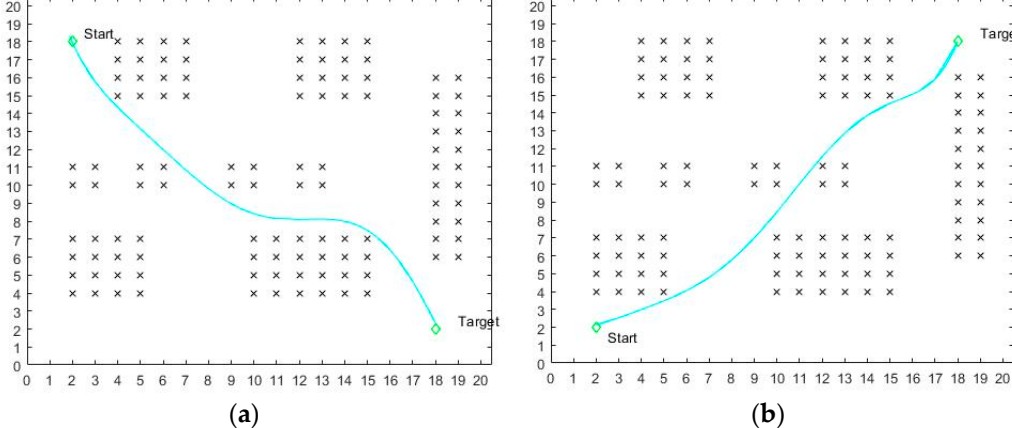

**Figure 12.** Path fitting results. (**a**) Figure 11a path fit; (**b**) Figure 11b path fit.

As shown in Figure 11, the path obtained by the artificial potential field method is discontinuous in the global path steering (see Section 3.2.2 above for the rationale). We, therefore, used the least squares method of path fitting to obtain Figure 12. Figure 12 gives the fitted paths planned by the algorithm based on different starting and ending points in the same environment.

In the case of local path planning, the repulsive force from the obstacles is only applied to the moving car within a certain range with the moving car as the center of the circle, and the repulsive force from the obstacles outside the range is 0. In addition, considering that



dynamic obstacles are inevitable in the real environment, 20 random dynamic obstacles were included, indicated by the circles in Figure 13.

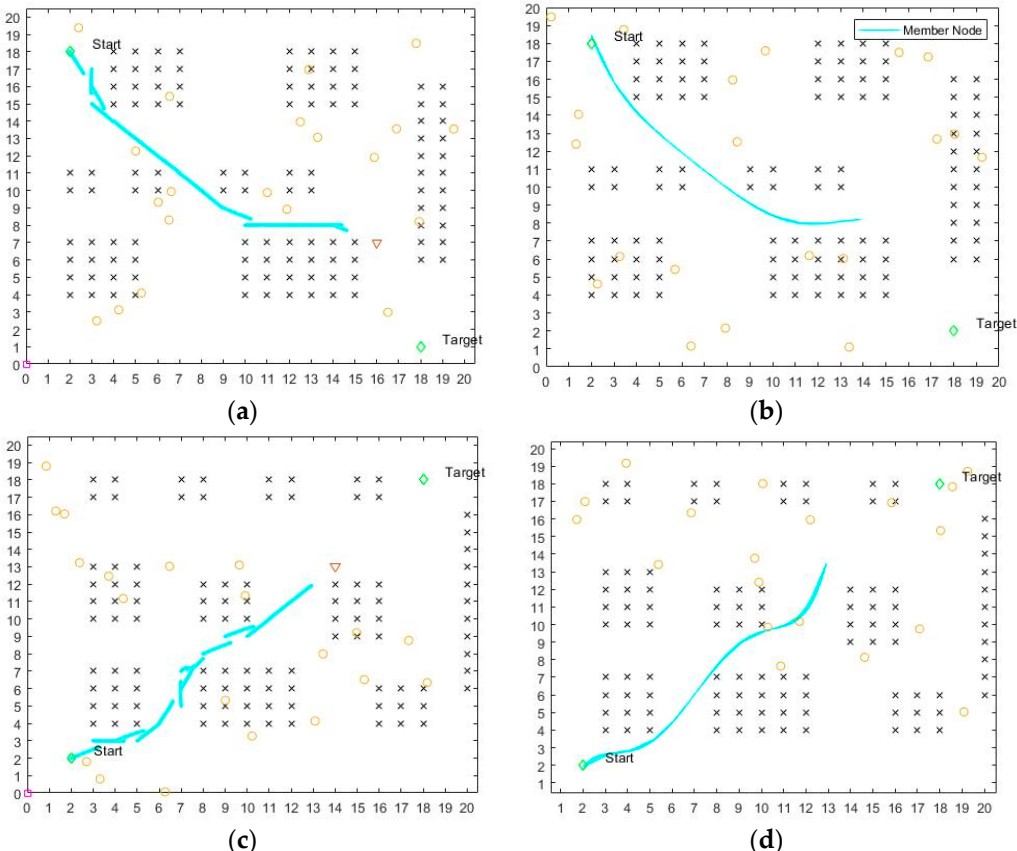

**Figure 13.** Addition of dynamic obstacles.

From the simulation results in Figure 13a, it can be seen that during the pathfinding process, when dynamic obstacles appear within the force range of the vehicle, the mobile robot will react quickly to avoid the obstacles. It is guaranteed to move towards the target point while moving towards the safe area as far as possible, as shown in Figure 13a at points (8, 9) and (12, 8). The vehicle based on global path planning can achieve real-time dynamic obstacle avoidance and reach the target point smoothly and safely. Meanwhile, as shown in Figure 13c,d, after the map environment is changed, the mobile robot path-planning process can still maintain the ability to quickly avoid dynamic obstacles while traveling towards the target point. As shown in Figure 13b, the vehicle can achieve real-time dynamic obstacle avoidance based on global path planning and reach the target point smoothly and safely.

### 4.2. Comparative Analysis of Optimized A-Star Algorithm and Bidirectional A-Star Algorithm

Currently, many scholars have proposed improving the bidirectional A-star algorithm for path planning [23–25]. The principle of the bidirectional A-star algorithm is to select a virtual endpoint in the middle of the straight line distance between the starting point and the ending point [26]. If the virtual endpoint lies in an obstacle area, the nearest obstacle edge is chosen as the virtual endpoint, while the endpoint at the other end is used as the starting point, and then path planning is performed towards the virtual endpoint.

As shown in Figure 14a,c, (10.9) is defined as the midpoint of the bidirectional A-star, indicated by the red "☆" in the diagram. A comparison of the path planning of the optimized A-star algorithm and the bidirectional A-star algorithm proposed in this paper shows that the optimized A-star algorithm has better throughput, that the path-planning efficiency of the optimized A-star algorithm is higher (as shown in Table 4), and that the

path-planning time of the optimized A-star algorithm is 65.2% less than the path-planning time of the bidirectional A-star algorithm. The number of computing nodes is 103 in Figure 14a and 70 in Figure 14b. The optimized A-star algorithm reduces the number of computing nodes by approximately 32% compared to the bidirectional A-star algorithm. Meanwhile, as shown in Figure 14a,c, the bidirectional A-star algorithm is affected by various factors such as obstacle size and map environment complexity when selecting virtual endpoints, which indirectly affects the pathfinding efficiency of the bidirectional A-star algorithm. The pathfinding efficiency of the optimized A-star algorithm is only affected by the complexity of the map environment, and therefore the performance of the optimized A-star algorithm is more stable than that of the bidirectional A-star algorithm.

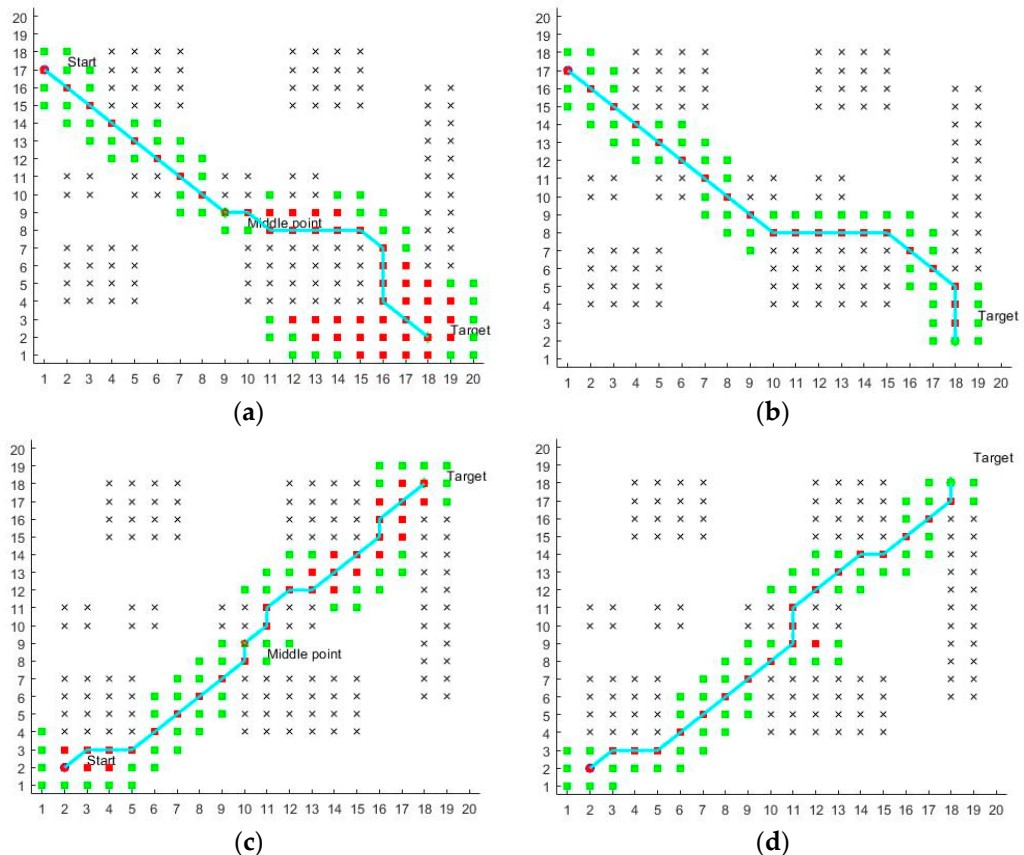

**Figure 14.** Optimized A-star algorithm vs. the bidirectional A-star algorithm. (**a,c**) The bidirectional A-star algorithm; (**b,d**) optimization of the A-star algorithm.

**Table 4.** Comparison of path-planning times based on Figure 14a,b (unit: s).

| Time | 1 | 2 | 3 | 4 | 5 | 6 | 7 | 8 | 9 | 10 |
|---|---|---|---|---|---|---|---|---|---|---|
| Two-way exploration A-star algorithm | 0.486 | 0.493 | 0.497 | 0.452 | 0.462 | 0.489 | 0.473 | 0.472 | 0.458 | 0.454 |
| Optimization of the A-star algorithm | 0.163 | 0.157 | 0.161 | 0.158 | 0.154 | 0.162 | 0.169 | 0.181 | 0.183 | 0.158 |

*4.3. Simulation Analysis of the Effect of Different L Values on the Potential Field Method*

In this paper, the algorithm rules for path planning in the optimized potential field method are changed, replacing the fixed starting point and fixed endpoint of the traditional potential field method with temporary starting points and temporary endpoints that change as the position of the mobile robot changes. As a result, the number of iterations of the traditional potential field method is no longer suitable for the optimized potential field method. For this reason, an adaptive iteration number setting is proposed in this paper,

which has been theoretically derived in Section 3.2.1. The experimental part was set up
with L values of 1, 10 and 100 for comparison, and the results of the comparison are shown
below.

From Figure 15a,b, it can be seen that when the value of L is too small, it leads to
too few iterations, and therefore it is difficult for the optimized potential field method to
reach the interim endpoint. In addition, as shown in Figure 16, the path-planning time
of the optimized potential field method is only reduced by 30% when the value of L is
1 compared to when the value of L is 10. From Figure 15b,c, the path-planning results
are almost the same for L values of 10 and 100. However, as can be seen in Figure 16, the
optimized potential field method time increases by 288% for the L value of 100 compared
to the L value of 10. Therefore, it can be concluded that when the L value is too large, it
increases the path-planning time of the optimized potential field method significantly, but
there is no significant improvement in the final path-planning result.

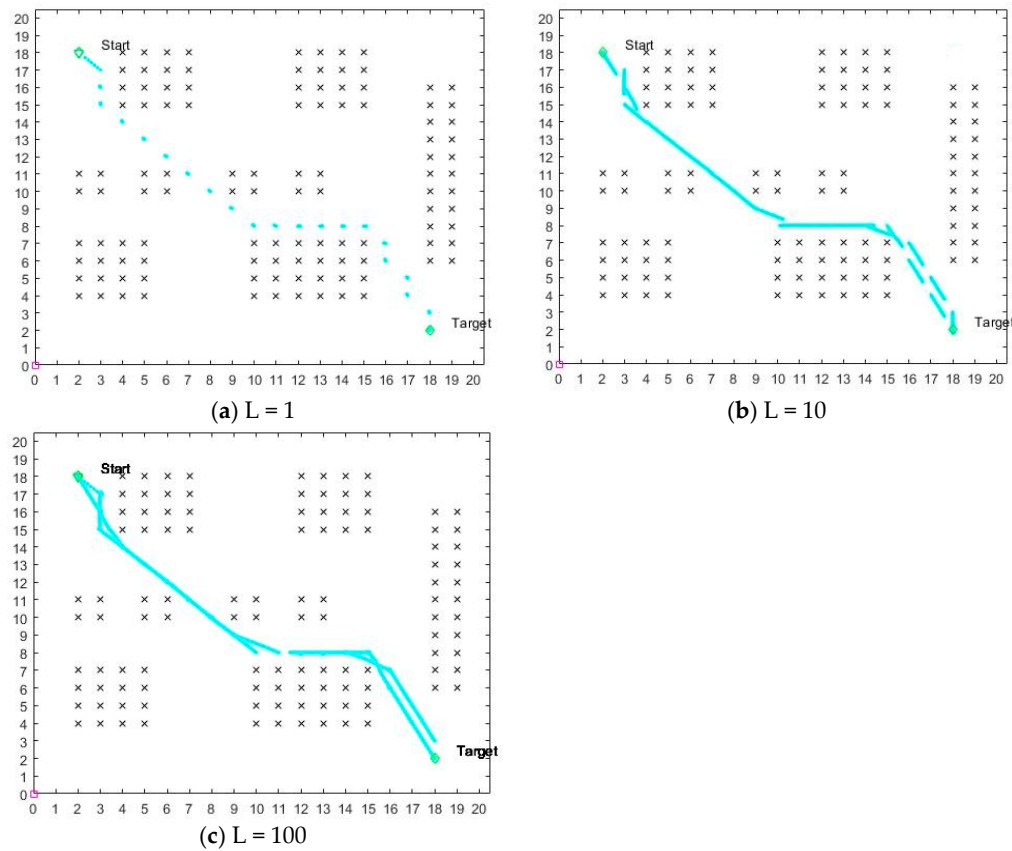

(**a**) L = 1

(**b**) L = 10

(**c**) L = 100

**Figure 15.** Effect on the optimized potential field method for different L values.

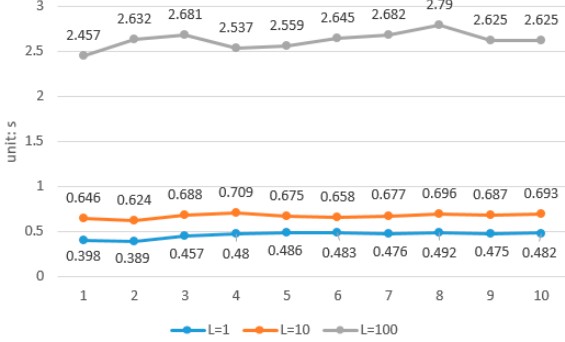

**Figure 16.** Optimal potential field method path-planning time for three L values.

### 4.4. Comparative Analysis of Fusion and Ant Colony Algorithms

The principle of the ant colony algorithm represents the feasible solution to the problem to be optimized in terms of the paths taken by ants, with all the paths of the entire ant colony forming the solution space of the problem to be optimized. The ants with shorter paths release more pheromones, and as time passes, the concentration of pheromones accumulated on the shorter paths gradually increases, and the number of ants choosing that path increases. Eventually, the entire ant population will concentrate on the best path under the effect of positive feedback, which then corresponds to the optimal solution of the problem to be optimized [27,28].

A comparison of the path-planning times from Figures 11b and 17b is shown in Table 5. The average planning time of the fusion algorithm is 0.722 s, and the average path-planning time of the ant colony algorithm is 2.073 s. The path-planning time of the fusion algorithm is reduced by 65.2% relative to the ant colony algorithm, which indicates that the fusion algorithm and the ant colony algorithm have the same path-planning requirements while completing the same path. This indicates that the fusion algorithm and the ant colony algorithm have the same path-planning time advantage while meeting the same path-planning requirements.

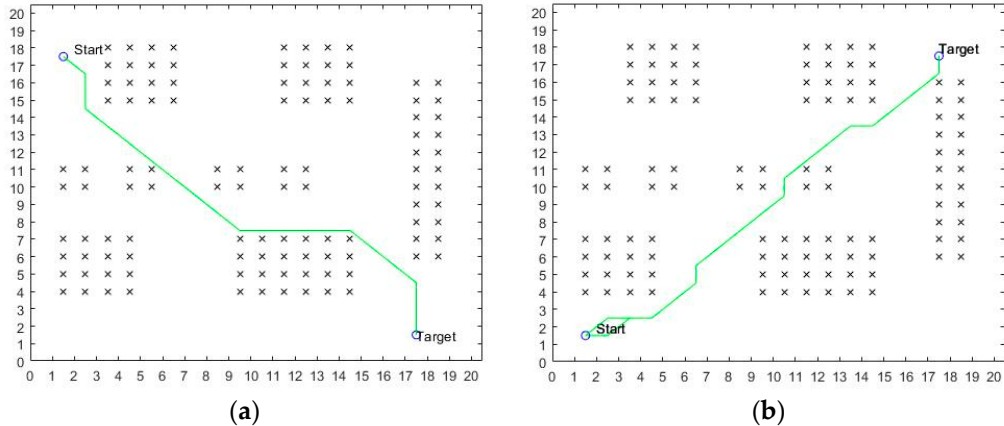

(**a**)                    (**b**)

**Figure 17.** Ant colony algorithm path planning.

**Table 5.** Comparison of path-planning time based on Figures 11b and 17b (unit: s).

| Time | 1 | 2 | 3 | 4 | 5 | 6 | 7 | 8 | 9 | 10 |
|---|---|---|---|---|---|---|---|---|---|---|
| Ant colony algorithm | 2.185 | 2.092 | 2.029 | 2.047 | 2.053 | 2.064 | 2.066 | 2.049 | 2.055 | 2.092 |
| Fusion algorithm | 0.693 | 0.712 | 0.706 | 0.718 | 0.700 | 0.750 | 0.740 | 0.736 | 0.738 | 0.726 |

### 4.5. Comparative Analysis of Fusion Algorithms and the RRT Algorithm

The RRT algorithm takes the starting point as the root node and adds leaf nodes by random sampling to generate a randomly expanded tree that is able to find a path from the starting point to the target point when the target point lies on the randomly expanded tree [29,30].

A comparison of the path-planning times for Figures 11a and 18a is shown in Table 6, with the same guaranteed obstacle environment and the same start and end points. The red circled area indicates the target point area. When the path is planned to this area, the path planning is completed. The average planning time of the fusion algorithm is 0.655 s. The average path-planning time of the RRT algorithm is 4.003 s. The path-planning time of the fusion algorithm is reduced by 83.64% relative to the path-planning time of the RRT algorithm. This indicates that the fusion algorithm and the RRT algorithm have a greater path-planning time advantage while meeting the same path-planning requirements.

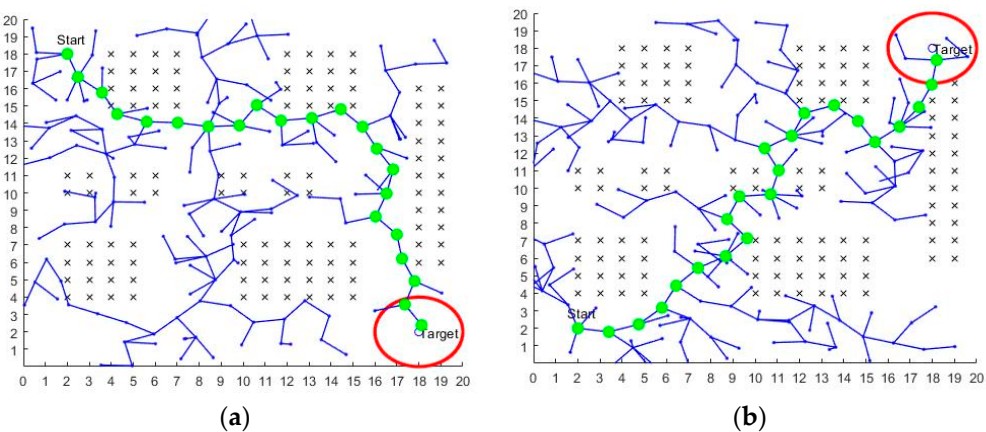

(**a**)  (**b**)

**Figure 18.** The RRT algorithm path planning.

**Table 6.** Comparison of path-planning time based on Figures 11a and 18a (unit: s).

| Time | 1 | 2 | 3 | 4 | 5 | 6 | 7 | 8 | 9 | 10 |
|---|---|---|---|---|---|---|---|---|---|---|
| The RRT algorithm | 3.779 | 3.871 | 3.996 | 4.063 | 3.921 | 4.012 | 4.039 | 4.182 | 4.094 | 4.074 |
| Fusion algorithm | 0.593 | 0.633 | 0.677 | 0.715 | 0.640 | 0.647 | 0.658 | 0.648 | 0.680 | 0.660 |

## 5. Results and Discussion

Due to the increasingly complex obstacle environments faced by mobile robots, it has been difficult for traditional path-planning algorithms to meet the path-planning needs of mobile robots, so this paper proposes a path-planning algorithm that incorporates the optimized A-star algorithm and the artificial potential field method. For the traditional A-star algorithm, as of now, many scholars are choosing to incorporate function constraints or implement algorithmic parallel pathfinding methods such as the bidirectional A-star algorithm. Such optimization provides very limited performance improvement to the traditional A-star algorithm, and a comparative analysis is also presented in the simulation section of this paper. Therefore, future optimization directions for the A-star algorithm should be able to significantly reduce the amount of algorithmic computation and increase path optimality.

The artificial potential field method has the problem of not being able to perform optimization and easily falling into local optimality, so the artificial potential field method can complete path planning but not necessarily find the optimal path. This leads to the fact that the artificial potential field method is capable of path planning, but may not necessarily find the optimal path, or may fall into a local optimum at some location in the map. This is an important issue in the current study of artificial potential field methods for path planning. This paper, therefore, uses the data obtained from the global path planning of the optimized A-star algorithm to optimize the artificial potential field method, thus limiting the path-planning space of the artificial potential field method. The problem of the artificial potential field method tending to fall into local optimality and path non-optimality is thus solved.

Each algorithm has its own strengths and weaknesses and limitations in the use of scenarios. In order to achieve complementary advantages, researchers at home and abroad in recent years have preferred to fuse multiple algorithms, which will also be the development direction for path planning for a long time.

## 6. Conclusions

In this paper, the traditional A-star algorithm is optimized, and new pathfinding rules and algorithm structures are designed to obtain global path-planning information. The data obtained by the optimized A-star algorithm are applied to local path planning, a new pathfinding rule for the potential field method is proposed, an intermittent point

pathfinding strategy is added and an intermittent point judgment function is constructed. The results show that the fusion algorithm can reduce the pathfinding time by about 40% while guaranteeing the same path as the traditional algorithm. The fusion algorithm also reduces the probability of the potential field method falling into local optima, improves the smoothness of the planned path at the turn, and enables the robot to find a safe path quickly even in complex environments. In this paper, the fusion-optimized A-star algorithm is compared with the more advanced bidirectional A-star algorithm, the ant colony algorithm and the RRT algorithm for path-planning time to demonstrate the advanced nature of the optimization algorithm. Through a series of experiments and data analyses, it can be concluded that the fusion algorithm in this paper can effectively improve the path-planning capability of mobile robots in complex scenarios, but its operation speed still needs to be improved, which is the focus of the next step.

**Author Contributions:** All of the authors contributed extensively to the work. B.W. proposed the key ideas, analyzed the key contents using a simulation and wrote the manuscript; L.L. obtained the financial support for the project leading to this publication; H.X. modified the manuscript. All authors have read and agreed to the published version of the manuscript.

**Funding:** This work was supported in part by the Initial Scientific Research Fund of FJUT under Grant GY-Z12079, Grant GY-Z21036 and Grant GY-Z20067 and in part by Fujian Provincial Science and Technology Department (Grant 2022H6005 and Grant 2022J01952).

**Acknowledgments:** The authors would like to thank the anonymous reviewers for their valuable comments.

**Conflicts of Interest:** The authors declare no conflict of interest.

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
