# Peer review of "Research on Path-Planning Algorithm Integrating Optimization A-Star Algorithm and Artificial Potential Field Method"

_electronics, doi:10.3390/electronics11223660_

Round 1

Reviewer 1 Report

The motivation and background behind the work are solid and I congratulate the authors for this. However, there are many places that need improvement. 

For example:

The main contribution of Section 2.2 Optimization of the A-star algorithm, is not clearly explained. Line 136, 137 are not clear especially with the use of terms such as evaluation function (you can refer to a particular equation) or the use of phrase "in situ". 

Definitions of parent node is not clear. Definitions of F-value, G-value, Open-list, close list are not clear and have to come before the algorithm is presented.

Optimization of potential field method section is also not clearly presented. What is the "turning point of global path"? What are "Judgement functions"?

Why is the coordinate AFTER current point named c-1 and coordinate BEFORE current point named c+1? Looks very odd and counter-intuitive.

Iteration parameter is not clear. Where is it being used?

A final algorithm or block diagram to understand the overall architecture of planning would be very useful. 

Reviewer 2 Report

Article 1 Research on path planning algorithm integrating optimization 2 A-star algorithm and artificial potential field method.

Comment1: The authors ompared  traditional A-star al- 21 gorithm and the bidirectional A-star algorithm, the ant colony algorithm and the RRT algorithm. Numerical results of comparision  may be written in abstract .

Comment2: Classification of path planning algorithm (figure ) may introduce in introduction

Comment3: Figure 3. Force analysis of robot in artificial potential field visibility should improve.

Comment4: Figure 6.  Global path comparison parameter should be written on x axis and y axis and other graph too.

Comment5: introduce  Results and discussion section before Conclusion.

Reviewer 3 Report

In this paper, the author proposes a path planning algorithm for the operation of a cleaning robot. The proposed algorithm solves the computational problem of A* and the local minima of the potential field.

1. The 'new' expression is a powerful expression for an algorithm that has never been tried before.

As introduced in 'Path Planning Using Artificial Dislocation Method and A-Star Convergence Algorithm' in 2022, the method proposed by the author is not a previous attempt. Therefore, to avoid misunderstanding by the reader, other expressions should be used instead of strong ones.

2. The effect on the outcome should in theory be accounted for for some arbitrarily defined variable. For example, set L to 10 and explain and show experimentally how the variable affects the outcome.

3. Authors should be able to express algorithms in pseudocode so that readers can understand the conditions and flow of each algorithm.

4. There is no association between A-star and potential fields. In the current description, it seems that the weakness of the potential field has been supplemented with the A-star algorithm rather than the fusion.

5. Since it is intended for indoor cleaning robots, it is necessary to include experimental results at a level applicable to cleaning robots. It works stably even in an enclosed space such as indoors, and please attach the result of reduced computation.

Author Response

请参阅附件。

Reviewer 4 Report

## Reviewer

- Confidence: 3/5

## Title 

This paper presents fusion algorithm to improve path planning based on A-star algorithm and artificial potential field method. 

The proposed approach then validated using MATLAB simulations.

## Summary

Major point to be considered:

1) In Section 4 (Simulation and Analysis), Figures title and its corresponding Table or explaination is confusing, e.g., 

What is Subfigure a~g in Figure 6? The following Table 1 is also not clear, i.e., which one is Figure 6 (a), Traditional or Optimized algorithm ?

I suggest to check ALL figures and subfigures title in this section and check the corresponding Table or explaination to improve overall readibility.

Point to be considered:

- p7 248: I found ... 

- p10 324: Compared with the traditional a-star (small a or big a ? please check for overall consistency)

- p14: The red "*" is the virtual end point  

Author Response

请参阅附件。

Round 2

Reviewer 2 Report

Authors response satisfactory